



# Linear Variation of M₂ Tide in the East and South China Seas

Hao Ke[1,2], Yixin Lu[3], Jian Wang[1,2], Weifeng Hao[1,2], Tianhao Ding[1,2]

[1]Chinese Antarctic Center of Surveying and Mapping, Wuhan University, Wuhan 430072, China
[2]Key Laboratory of Polar Environment Monitoring and Public Governance (Wuhan University), Ministry of Education, Wuhan 430072, China
[3]School of Resource and Environmental Sciences, Wuhan University, 430079, China

*Correspondence to*: Hao Ke (kehao1984@whu.edu.cn)

**Abstract.** Considering the time-varying tidal parameters in offshore China, we proposed a harmonic analysis method combined with an additional time-varying mode. Tidal analysis was performed on nearly 20 years of sea surface altimetry data from the Jason 1-3, Envisat, and ERS-2 satellites, and 13 tide stations located along the coast of the southeast China Sea. The results show that there are several peak regions for the amplitude and phase lag change rate of the M₂ tide in the East China Sea and the South China Sea, which are mainly located in the estuary areas of inland river basins, such as the Yangtze Estuary, the Pearl River Estuary, and the Nanliu Estuary, and the variation ranges are roughly 2.0–4.0 mm/yr and 0.8–2.0 ⁰/yr. The variations in tidal parameters were attributed to the changes in water depth and coastline in the estuaries.

## 1 Introduction

Tidal movement is one of the main forms of dynamic processes in the ocean (Fang et al., 2004), and its motion is mainly represented by the harmonic constants of the tide. By analyzing the spatial trend distribution of the major tidal components, linear variations were found to be common along the North American coast (Woodworth, 2010; Müller, 2011). In addition, linear trends in the amplitude and phase of tidal constituents have been found around the British Isles, the southern North Sea, the Gulf of Maine, Northwest America, the North Atlantic Ocean, the Caribbean Sea, and the East China Sea (Amin, 1983; Ray, 2006; Jay, 2009; Torres & Tsimplis, 2011; Feng & Tsimplis, 2014).

The Chinese coasts have been dramatically altered by sediment deposition in estuaries (Chu et al., 2006; Pelling et al., 2013), coastal erosion under the influence of frequent tropical cyclones (Qi et al., 2010; Feng et al., 2012), and extensive reclamation projects constructed over the past few decades, all of which may potentially alter the offshore tidal characteristics (Li et al., 2004; Li et al., 2013; Wang et al., 2014; Wang et al., 2011).

Based on the tidal analysis of the water level data observed from 1954 to 2012 at 17 tidal gauges in the Bohai Sea, Yellow Sea, North Sea, and Taiwan Strait of China, the tidal constituents exhibit trends of long-term change. The amplitudes of M₂ tide constituents in the Bohai Sea and Yellow Sea exhibit maximum increases of 4–7 mm/yr, and the most significant rate





reported is 7 mm/yr for the $M_2$ amplitude occurring at Lianyungang on the west coast of the Yellow Sea (Feng & Tsimplis,

2014). In the Taiwan Strait, it reached 0.3–1.4 mm/yr (Feng et al., 2015).

It appears that the linear variation in tidal parameters is objective and significant. However, most of the studies on the

changes in tidal parameters have focused on the data obtained from tidal gauges along the continental coast, and the research

on larger areas of offshore waters is rare. To address this, we used satellite altimetry data and tide gauges observation data to

capture the changes in tidal parameters in the East and South China Seas. Moreover, considering the sparse time sampling

rate of satellite altimetry data, the analysis method of tidal variation at tide gauges may not be applicable. Therefore, we

proposed an additional time-varying mode for the harmonic analysis of tide changes.

The remainder of this article is organized as follows. Section 2 provides a detailed description of the proposed method. The

linear variations of $M_2$ in the China Seas are discussed in Section 3. Finally, the conclusion and future research directions are

presented in Section 4.

**2 Methods**

The harmonic analysis method with an additional time-varying mode assumes that the mean sea level, the amplitude, and the

phase lag in the tide height equation all change with time, as do the nodal correction parameters for the amplitude and phase.

The tide height equation can be expressed as follows:

$$h(t) = S(t) + \sum_{i=1}^{N} f(t)_i H(t)_i \cos[V(t)_i + u(t)_i - g(t)_i] \tag{1}$$

Where $h(t)$ indicates the sequence of water surface height, $S(t)$ is the elevation sequence of mean sea level, $V(t)$ is the phase

of each tide constituent at the time $t$, which varies uniformly over time. And $H(t)$ and $g(t)$ are the sequences of tide

amplitudes and phase lags, respectively. In general, the tidal harmonic constants, namely the amplitude and phase lag, are

constant, but in this study we assume that they also change with time, therefore, they are set to $H(t)$ and $g(t)$. Then, $f(t)$ and

$u(t)$ are the nodal correction parameters for the amplitude and phase, respectively. $N$ represents the total number of tide

constituents, usually taken as 13 according to the specifications for hydrographic survey GB 12327-2022. These 13 tide

constituents consist of 4 semi-diurnal $M_2$, $S_2$, $N_2$, $K_2$, and 4 diurnal $K_1$, $O_1$, $P_1$, $Q_1$, and 3 shallow tide constituents $M_4$, $MS_4$,

$M_6$, and 2 long-period meteorological tide constituents Sa and Ssa.

Cosine expansion gives the following:

$$h(t) = S(t) + \sum_{i=1}^{N} \{f(t)_i \cos[V(t)_i + u(t)_i]H(t)_i \cos g(t)_i + f(t)_i \sin[V(t)_i + u(t)_i]H(t)_i \sin g(t)_i\} \tag{2}$$
$$= S(t) + \sum_{i=1}^{N} [a(t)_i H(t)_i \cos g(t)_i + b(t)_i H(t)_i \sin g(t)_i]$$





Where $a(t)_i = f(t)_i \cos[V(t)_i + u(t)_i]; b(t)_i = f(t)_i \sin[V(t)_i + u(t)_i]$, and the values of coefficients $a$ and $b$ can be calculated at each time $t$.

It is assumed that the sine and cosine components of tide constituents change slowly like the mean sea level changes over a long period of time, then:

$$S(t) = S_0 + dS = S_0 + \dot{S} \cdot \Delta t$$

$$H(t)_i \cos g(t)_i = H_{0i} \cos g_{0i} + d(H \cos g)_i = HC_{0i} + \dot{HC}_i \Delta t \qquad (3)$$

$$H(t)_i \sin g(t)_i = H_{0i} \sin g_{0i} + d(H \sin g)_i = HS_{0i} + \dot{HS}_i \Delta t$$

In Equation (3), $S_0$ and $\dot{S}$ are the initial value and linear change rate of the mean sea level, respectively. Similarly, $HC_0$ and $HS_0$ represent the initial values for the cosine and sine components, and $\dot{HC}_i$ and $\dot{HS}_i$ are the change rates, respectively. $\Delta t$ is the time difference between time $t$ and start time $t_0$. According to Equations (2) and (3), the least square adjustment is adopted, and the coefficient matrix, unknown parameters matrix, and observation matrix are listed as follows.

$$B = \begin{bmatrix} 1 & \Delta t^1 & a_1^1 & a_1^1 \cdot \Delta t^1 & b_1^1 & b_1^1 \cdot \Delta t^1 & \cdots & a_{13}^1 & a_{13}^1 \cdot \Delta t^1 & b_{13}^1 & b_{13}^1 \cdot \Delta t^1 \\ 1 & \Delta t^2 & a_1^2 & a_1^2 \cdot \Delta t^2 & b_1^2 & b_1^2 \cdot \Delta t^2 & \cdots & a_{13}^2 & a_{13}^2 \cdot \Delta t^2 & b_{13}^2 & b_{13}^2 \cdot \Delta t^2 \\ \vdots & \vdots & \vdots & \vdots & \vdots & \vdots & \ddots & \vdots & \vdots & \vdots & \vdots \\ 1 & \Delta t^{m-1} & a_1^{m-1} & a_1^{m-1} \cdot \Delta t^{m-1} & b_1^{m-1} & b_1^{m-1} \cdot \Delta t^{m-1} & \cdots & a_{13}^{m-1} & a_{13}^{m-1} \cdot \Delta t^{m-1} & b_{13}^{m-1} & b_{13}^{m-1} \cdot \Delta t^{m-1} \\ 1 & \Delta t^m & a_1^m & a_1^m \cdot \Delta t^m & b_1^m & b_1^m \cdot \Delta t^m & \cdots & a_{13}^m & a_{13}^m \cdot \Delta t^m & b_{13}^m & b_{13}^m \cdot \Delta t^m \end{bmatrix}$$

$$X = \begin{bmatrix} S_0 & \dot{S}_0 & HC_1^0 & \dot{HC}_1 & HS_1^0 & \dot{HS}_1 & \cdots & HC_{13}^0 & \dot{HC}_{13} & HS_{13}^0 & \dot{HS}_{13} \end{bmatrix}^T$$

$$h = [h(\Delta t^1) \quad h(\Delta t^2) \quad \cdots \quad h(\Delta t^m)]^T \qquad (4)$$

The final solution of $X$ can be solved as follows:

$$X = (B^T P B) B^T P h \qquad (5)$$

The tidal analysis uses 13 tide constituents, namely $M_2$, $S_2$, $N_0$, $K_2$, $K_1$, $O_1$, $P_1$, $Q_1$, $M_4$, $MS_4$, $M_6$, $S_a$, and $S_{Sa}$. According to Equation (5), the annual change rates of the tide constituents can be obtained. Taking the $M_2$ tide component as an example, the time series expressions of the cosine and sine components are established according to the following equations:

$$HC_{M2}(\Delta t) = HC_{M2}^0 + \Delta t \cdot \dot{HC}_{M2}$$

$$HS_{M2}(\Delta t) = HS_{M2}^0 + \Delta t \cdot \dot{HS}_{M2} \qquad (6)$$

Then, the time series of the $M_2$ tidal amplitude and phase lag are solved:





$$H_{M2}(\Delta t) = \sqrt{HC_{M2}(\Delta t)^2 + HS_{M2}(\Delta t)^2}$$
$$g_{M2}(\Delta t) = \arctan[HS_{M2}(\Delta t) / HC_{M2}(\Delta t)]$$

(7)

At this point, the linear term of $H_{M2}$ and $g_{M2}$ can be obtained according to linear regression analysis. The specific calculation principle is as follows:

$$H_{M2}(\Delta t) = H_{M2}^0 + \Delta t \bullet \dot{H}_{M2}$$
$$g_{M2}(\Delta t) = g_{M2}^0 + \Delta t \bullet \dot{g}_{M2}$$

(8)

In the Equation (8), $H_{M2}^0$ and $g_{M2}^0$ are the initial values of $M_2$ tide component, respectively. $\dot{H}_{M2}$ and $\dot{g}_{M2}$ are the results that are the desired change rates of $M_2$ amplitude and phase lag.

## 3 Data

### 3.1 Satellite altimetry data

The Jason series satellite altimetry data derived from January 2002 to December 2020 have been adopted, whose repeated observation period is approximately 9.9 d. The longest synodic period is 9.19 yrs for $P_1$ and $K_2$ tide components according to the Rayleigh criterion. Therefore, nearly 20 years of observation can meet the requirement of tidal separation. In addition, ERS-2 and Envisat altimetry data, covering May 1995 to October 2010, have also been obtained for analysis. Because the repeated observation period is 35 d, which is an integral multiple of the period of the solar component $S_2$, this component can not be separated out. Similarly, $P_1$ and $K_1$ are also difficult to separate due to their close aliasing periods. However, for the $M_2$, the longest synodic period is 8.7 years between $M_2$ and $N_2$, so the observation data of 15 years derived from ERS-2 and Envisat can also meet the conditions for accurately separating the $M_2$ component. Table 1 shows the observation period of each satellite.

**Table 1. Starting time and ending time for each satellite observation period**

| Satellite | Starting time(UTC) | Ending time(UTC) |
|---|---|---|
| ERS-2 | 1995-05-14 01:33:35 | 2003-07-02 07:15:16 |
| ENVISAT | 2002-05-14 18:32:32 | 2010-10-18 21:34:01 |
| Jason-1(cycle001-240) | 2002-01-15 06:07:06 | 2008-07-21 23:17:41 |
| Jason-2(cycle000-280) | 2008-07-04 11:34:26 | 2016-02-17 10:27:25 |
| Jason-3(cycle000-179) | 2016-02-12 01:11:09 | 2020-12-27 08:04:55 |





### 3.2 Tide gauge observations

The tide observation data are derived from the University of Hawaii Sea Level Center and the Hydrology Bureau of the Yangtze River Water Resources Commission of China. All the tide gauges are located along the coast of the mainland and Taiwan, as shown in Figure 1. Table 2 shows the location and observation period of each tide gauge. The sampling interval of water level observation at all tide gauges is 1 h. The shortest observation time period was 6 years, and the longest was 44 years, which can meet the time requirements of tidal analysis.

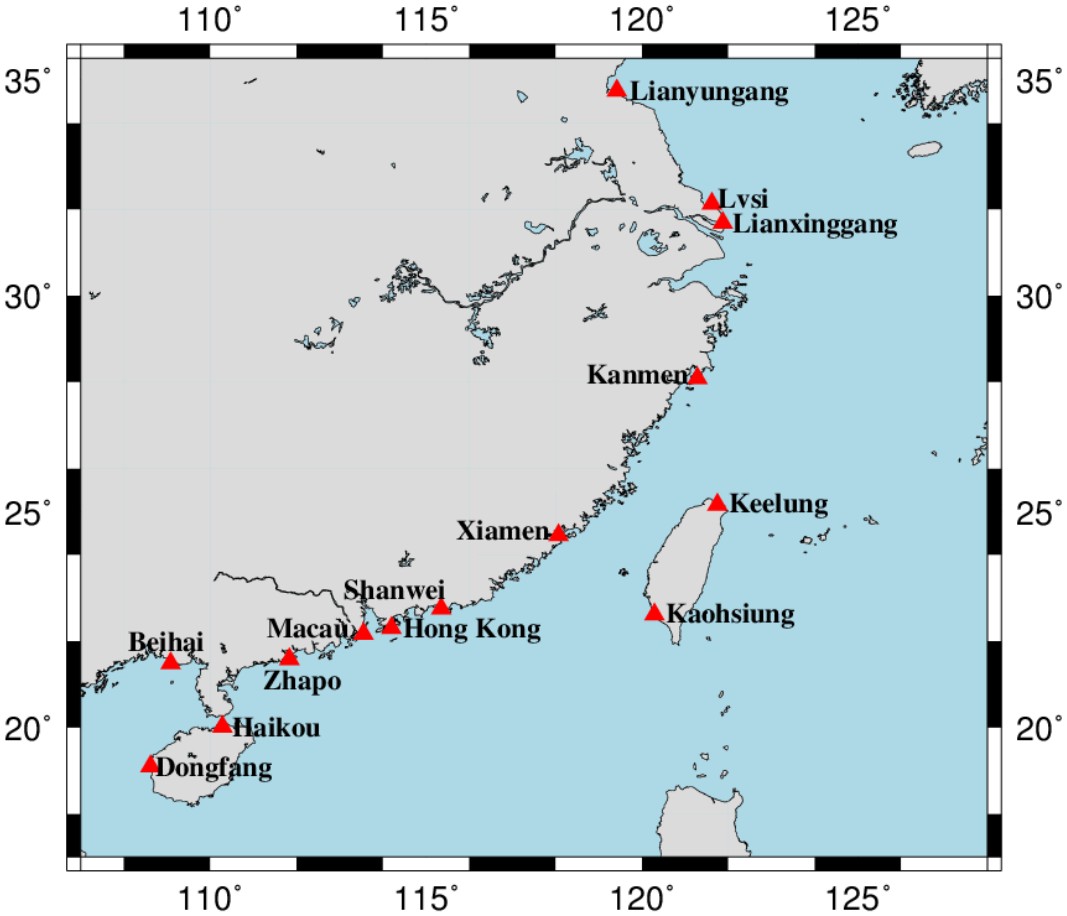

**Figure 1: Distribution of tide gauges in the China Sea**

**Table 2. Location, starting time, and ending time for each tide gauge observation period**

| Tide Gauge | Longitude (°) | Latitude (°) | Observation period |
|---|---|---|---|
| Xiamen | 118.067 | 24.450 | 1954.01–1997.12 |
| Kaohsiung | 120.288 | 22.615 | 1980.01–2016.12 |
| Keelung | 121.745 | 25.157 | 1980.01–2016.12 |
| Hong Kong | 114.200 | 22.300 | 1962.01–2018.12 |





| Haikou | 110.283 | 20.017 | 1976.01–1997.12 |
| Beihai | 109.083 | 21.483 | 1975.01–1997.12 |
| Zhapo | 111.833 | 21.583 | 1975.01–1997.12 |
| Macau | 113.550 | 22.167 | 1978.01–1985.12 |
| Shanwei | 115.350 | 22.750 | 1975.01–1997.12 |
| Dongfang | 108.617 | 19.100 | 1975.01–1997.12 |
| Lianyungang | 119.417 | 34.750 | 1975.01–1997.12 |
| Kanmen | 121.282 | 28.088 | 1975.01–1997.12 |
| Lvsi | 121.617 | 32.133 | 1975.01–1996.10 |
| Lianxinggang | 121.872 | 31.686 | 2002.01–2007.12 |

## 4 Implementation and Results

For the tide gauges, the tidal harmonic analysis was implemented year by year, using one year as the unit to obtain the time
series of the main tide constituent harmonic constants. In addition, linear regression analysis was used to calculate the linear
change rate of the amplitude and phase lag. Table 3 shows the linear variation of $M_2$ tide component at each tide gauge.

**Table 3. Linear variation of M2 for each tide gauge**

| TG | $M_2$ amplitude | | $M_2$ phase lag | |
|---|---|---|---|---|
| | $H$(cm) | $H$_rate(mm/yr) | $g$(°) | $g$_rate(°/yr) |
| Xiamen | 183.7 | 1.6 | 120.16 | -0.056 |
| Kaohsiung | 17.9 | 0.3 | 349.66 | 0.062 |
| Keelung | 22.4 | 0.8 | 46.98 | -0.036 |
| Hong Kong | 39.1 | -0.3 | 7.19 | 0.021 |
| Haikou | 22.3 | 0.3 | 26.27 | -0.061 |
| Beihai | 45.9 | 0.6 | 307.0 | 0.003 |
| Zhapo | 64.8 | 0.0 | 61.34 | -0.031 |
| Macau | 46.2 | -1.1 | 56.40 | 0.086 |
| Shanwei | 27.7 | -0.05 | 23.08 | 0.109 |
| Dongfang | 18.2 | 0.2 | 189.12 | 0.078 |
| Lianyungang | 164.1 | 7.8 | 305.72 | 0.178 |
| Kanmen | 187.8 | 1.0 | 22.38 | -0.061 |
| Lvsi | 171.5 | 1.2 | 119.90 | -0.222 |
| Lianxinggang | 133.5 | 5.9 | 107.73 | 0.286 |

As shown in the table 3, the change rate of M2 amplitude at Lianyungang is the largest, reaching 7.8 mm/yr, which is close
to the result of 7 mm/yr (Feng & Tsimplis , 2014). Lianxinggang located at the estuary of the Yangtze River, hold the second
title, with an amplitude change rate of 5.9 mm/yr. On the whole, it can be found that the variation of $M_2$ amplitude in the
coastal waters of the East China Seas is much higher than that in the coastal waters of the South China Seas. In the tide





gauges to the south of Xiamen, the $M_2$ amplitude change rates are basically $\pm 1$ mm/yr, and the phase lag change rates are also basically $\pm 0.1$ °/yr.

The sea surface height series at the satellite altimeter ground track was first calculated using the satellite altimetry data. Then,
the change rates of $M_2$ were obtained by tidal harmonic analysis with an additional time-varying mode. In Figure 2, the linear variation of $M_2$ along the track in the East China Seas derived from Jason, ERS -2 and Envisat satellite altimetry were presented.

(a)

(b)

(c)

(d)

**Figure 2: (a) and (b) present the linear variation of amplitude and phase lag along the Jason track in the East China Seas. (c) and (d) present the the linear variation of amplitude and phase lag along the Envisat track in the East China Seas.**





The following Figure 3 shows the linear variation of $M_2$ along the satellite ground track in the South China Seas.

(a)

(b)

(c)

(d)

**Figure 3: (a) and (b) present the linear variation of amplitude and phase lag along the Jason track in the South China Seas. (c) and**
130 **(d) present the the linear variation of amplitude and phase lag along the Envisat track in the South China Seas.**

Finally, the calculation of inconsistent values at the ground crossing points was conducted; see Table 4 for details.

**Table 4. The inconsistent values and change rates of M2 in the East China and South China Seas. ECS, East China Sea; SCS, South China Sea.**

|  | Discrepancy mean | Root mean square | Discrepancy mean | Root mean square |
|---|---|---|---|---|
| $M_2$ ECS | Amplitude(cm) | | Phase(°) | |
| Jason | 2.16 | 5.51 | 0.66 | 0.78 |
| ERS-2/Envisat | 2.60 | 3.77 | 1.53 | 3.48 |
| CR of $M_2$ ECS | Amplitude rate(cm/yr) | | Phase rate(°/yr) | |
| Jason | 0.13 | 0.16 | 0.16 | 0.21 |



| ERS-2/Envisat | 0.35 | 0.52 | 0.36 | 0.49 |
|---|---|---|---|---|
| **M2**<br>**SCS** | **Amplitude(cm)** | | **Phase(°)** | |
| Jason | 4.35 | 12.79 | 8.23 | 14.55 |
| ERS-2/Envisat | 2.05 | 3.79 | 4.89 | 9.56 |
| **CR of M$_2$**<br>**SCS** | **Amplitude rate(cm/yr)** | | **Phase rate(°/yr)** | |
| Jason | 0.12 | 0.15 | 0.39 | 0.46 |
| ERS-2/Envisat | 0.32 | 0.45 | 0.74 | 1.03 |

135 The results in Table 4 show that the discrepancy mean of the amplitude change rate of $M_2$ derived of Jason in the East and South China Seas at the ground crossing points is about 1.2~1.3 mm/yr, and the root mean square is about 1.5~1.6 mm/yr. Then, the discrepancy mean of the phase lag change rate of $M_2$ at all ground crossing points is 0.16~0.39 °/yr, and the root mean square is 0.21~0.46 °/yr. However, the discrepancy mean and root mean square for $M_2$ change rate derived of ERS-2/Envisat have poor performance. The specific results are as follows: the discrepancy mean of the amplitude

140 change rate of $M_2$ at the all ground crossing points is about 3.2~3.5 mm/yr, and the root mean square is 4.5~5.2 mm/yr, while the discrepancy mean of the phase lag change rate is ahout 0.36~0.74 °/yr, and the root mean square is 0.49~1.03 °/yr. Based on the calculation of the change rates, Kriging interpolation and cubic spline smoothing methods were used to generate variation modes of $M_2$ over the East and South China Seas, as shown in Figure 4

In order to analyze the accuracy of the variation modes, the change rates of $M_2$ derived of tide gauge observations and modes

145 estimations have been listed in the Table 5. The circular colored dots in Figure 4 show the $M_2$ variation at the tide gauges. In the list of the tide gauges, Lianyungang station is actually beyond the research scope of $M_2$ variation mode construction, and it is located in the Yellow Sea of China. Therefore, the mode estimation at the Lianyungang belongs to extrapolation, the comparison results in Table 5 also verify the poor accuracy of extrapolation, in which the difference of the amplitude change rate of $M_2$ between tide gauge observation and mode estimation is 6.1 mm/yr, and the difference of the phase lag change rate

150 is 1.251 °/yr. In view of this poor performance, Lianyungang station has been excluded to evaluate the accuracy of $M_2$ variation modes.

Compared with $M_2$ variation modes, the mean differences in amplitude and phase lag are 1.71 mm/yr and 0.148 °/yr, and their corresponding standard deviations are 2.48 mm/yr and 0.184 °/yr, respectively. This set of accuracy statistics are worse than that of Jason's ground crossing points, but better than that of ERS-2/Envisat. The Jason series has a high measurement

155 accuracy of 2~3 cm, and a high observation sampling rate as about 10 days. The Jason observation data collected in the past 20 years can fully ensure the complete separation of the 13 tide constituents during the tidal harmonic analysis. Although the altimetry accuracy of ERS -2/Envisat is comparable to that of Jason, the $M_2$ can not be completely separated from the shallow tide constituent $MS_4$ because of the low sampling rate, even if there are about 15 years of observation data. As a kind of shallow tide constituent, $MS_4$ has a small amplitude. According the results of tidal harmonic analysis at the tide

160 gauges listed in the Table 2, the amplitudes of $MS_4$ are roughly 2~3 cm. We are convinced that this defect causes the





accuracy of the change rates obtained by ERS-2/Envisat to be poor than that of Jason. Consequently, when combining the change rates obtained by Jason and ERS-2/Envisat to generate $M_2$ variation modes, the accuracy of variation modes should theoretically be in between the two satellite results.

Generally, the tide gauges are usually located in the coastal waters of the mainland, and the accuracy of altimetry in these
165 places is poor. Therefore, the accuracy of the sea surface elevation obtained by satellite observation is also poor, which further leads to the low accuracy in tidal harmonic analysis. For the offshore waters, the situation is much better. The results in Table 5 show that the standard deviations of $M_2$ change rates by comparing the modes estimations and tide gauge observations are 2.48 mm/yr and 0.184 °/yr, respectively. These tide gauges are all distributed on the shore, so it is bold to assume that these accuracy results are the lower limit of the $M_2$ variation modes, however, due to the small number of tide
gauges, this hypothesis needs to be further verified.

**Table 5. The comparison of $M_2$ between variation modes and tide gauges.**

| Tide gauge | $M_2$ amplitude variation | | | $M_2$ phase lag variation | | |
|---|---|---|---|---|---|---|
| | TG_observation (mm/yr) | Mode estimate (mm/yr) | Difference (mm/yr) | TG_observation (°/yr) | Mode estimate (°/yr) | Difference (°/yr) |
| Xiamen | 1.6 | 0.5 | -1.1 | -0.056 | -0.051 | 0.005 |
| Kaohsiung | 0.3 | 0.4 | 0.1 | 0.062 | 0.069 | 0.007 |
| Keelung | 0.8 | 1.2 | 0.4 | -0.036 | -0.383 | -0.347 |
| Hong Kong | -0.3 | -1.2 | -0.9 | 0.021 | 0.253 | 0.232 |
| Haikou | 0.3 | 1.3 | 1.0 | -0.061 | -0.231 | -0.170 |
| Beihai | 0.6 | -7.0 | -7.6 | 0.003 | 0.006 | 0.003 |
| Zhapo | 0.0 | -0.1 | -0.1 | -0.031 | 0.111 | 0.142 |
| Macau | -1.1 | -3.6 | -2.5 | 0.086 | 0.215 | 0.129 |
| Shanwei | 0.0 | -0.7 | -0.7 | 0.109 | -0.291 | -0.400 |
| Dongfang | 0.2 | 3.9 | 3.7 | 0.078 | -0.067 | -0.145 |
| Lianyungan | 7.8 | 13.9 | 6.1 | 0.178 | -1.073 | -1.251 |
| Kanmen | 1.0 | 1.8 | 0.8 | -0.061 | -0.094 | -0.033 |
| Lvsi | 1.2 | 0.2 | -1.0 | -0.222 | -0.107 | 0.115 |
| Lianxingga | 5.9 | 3.6 | -2.3 | 0.286 | 0.091 | -0.195 |
| **M2 amplitude variation** | **Average of the absolute difference (mm/yr)** | | | **Standard deviation of difference (mm/yr)** | | |
| | 1.71 | | | 2.48 | | |
| **M2** | **Average of the absolute difference (°/yr)** | | | **Standard deviation of difference (°/yr)** | | |



| phase _lag variation | 0.148 | 0.184 |
|---|---|---|

In terms of the amplitude change rate, there are several peak regions in offshore waters along the coasts of the East China and South China Seas; specifically, located in the north of the East China Sea near Lianyungang, the waters near the mouth
of the Yangtze River, the north of the Taiwan Strait and the waters near Xiamen, the Pearl River Estuary of Hong Kong and Macao, and the Beihai waters of Guangxi. The amplitude change rate of $M_2$ in these regions is generally above 2 mm/yr. However, in the Pearl River Estuary and the North Sea near Hong Kong and Macao, the trend of variation in $M_2$ amplitude is opposite to that in the other sea areas mentioned above. The amplitude shows negative growth, and the change rate is approximately −4 mm/yr. Overall, the amplitude change rate is most remarkable at 200 to 300 km offshore, and then the
total rate decreases laterally, toward the shore and away from the shore.

Regarding phase lag, similar to amplitude variation, several peak areas are located in the north of the East China Sea near Lianyungang, the south of Xiamen, the Pearl River Estuary of Hong Kong and Macao, and the Beihai Sea of Guangxi. Among them, the change rate of phase lag in several peak areas of the South China Sea fluctuates sharply, from +0.8 °/yr to approximately −2.0°/yr. Overall, the change rate of phase lag in the South China Sea is much higher than that in the East
China Sea.

There are several reasons for the change in tidal parameters. However, the rapid change in water depth may be one of the main causes. Numerical simulations show that the coastline and water depth variation can significantly alter $M_2$ amplitude in the Bohai Sea and Yellow Sea (Pelling et al., 2013). China's rapid economic development in the middle and late last century, including massive tidal flat reclamation in coastal areas, created thousands of square kilometers of new land area and
significantly altered the coastline (Li et al., 2004; Li et al., 2013; Wang et al., 2014; Wang et al., 2011).

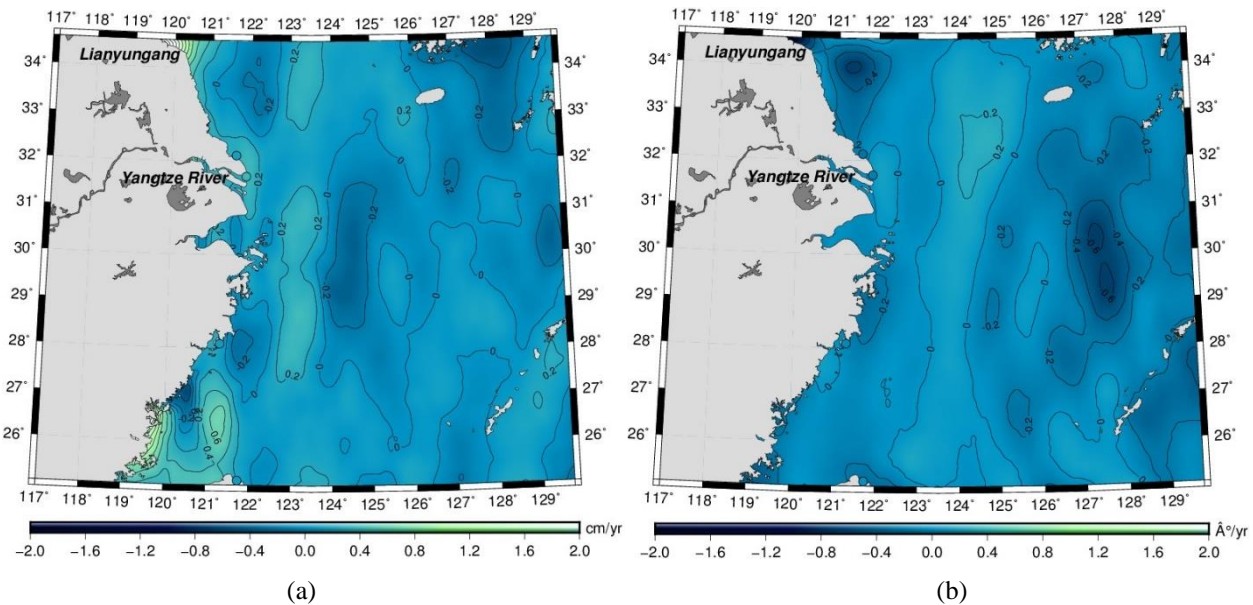

(a)                                                                (b)





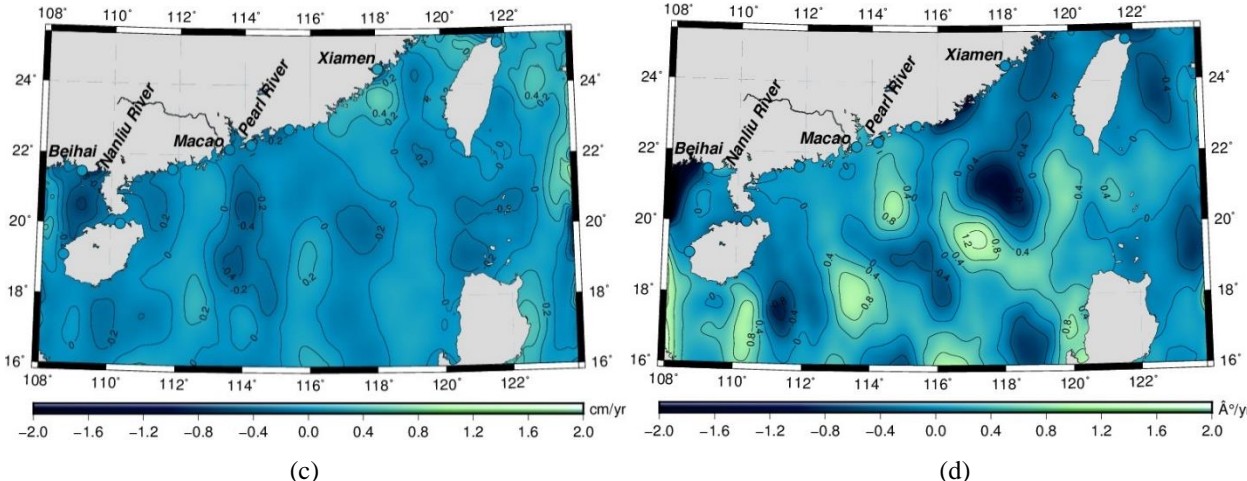

(c)                                                                      (d)

**Figure 4: (a) and (b) represent the change rates of amplitude and phase lag in the East China Sea, respectively. (c) and (d) correspond to change rates of amplitude and phase lag in the South China Sea, respectively.**

In the Yangtze and Pearl River mouths, the water depth was changed because of the large amount of sediment carried by the upper reaches (Zhang et al., 2008; Yang et al., 2011). Moreover, massive dam construction in the Yangtze River has resulted in channel erosion in the lower reaches and coarsening of bottom sediment, as well as erosion in the Yangtze River delta, which has not only changed the local water depth but also changed the coastline (Yang et al., 2011). Similarly, there is a Nanliu River Estuary in Beihai, and the delta coast used to have many branches that divided Dangjiang, Muan, Yujiang, Matou, and other places into islands. Since the 1950s, frequent changes were made in the channel, merging the islands into the land, and the waterway into the sea has been regulated into the mainstream. Furthermore, the East Channel, West Channel, Zhou River, and other branches have been cut off by dikes. These changes altered the local water depths and coastlines, causing dramatic changes in tidal parameters. From Figure 4, the amplitude and phase lag of $M_2$ in the Nanliu River, flowing into the Haikou area of the North Sea, are changing at a large rate.

**5 Conclusion**

In this paper, we proposed a harmonic analysis method that uses an additional time-varying mode and applied it to tidal analysis in the East China and South China Seas. The results showed that the tidal amplitude and phase lag of $M_2$ significantly changed in the East and South China Seas, and the change rates in the Yangtze River Estuary, Pearl River Estuary, and Nanliu Estuary exhibited the peak values. The main reason for the changes in tidal parameters was attributed to the changes in water depth and coastline in the estuaries. Other causes, such as the change in temperature and salinity in the estuary, and the influence of meteorological factors, such as typhoons and storm surges, should be considered more deeply in subsequent studies to determine their relative contributions.



**Acknowledgement**

This work was funded by National Natural Science Foundation of China under Grant 42171383, and funded by Key
Laboratory of Ocean Geomatics, Ministry of Natural Resources, China (Grant No. 2021B14).

 **Data Availability Statement**

The satellite altimetry data and water level observation data at tide gauges can be download from
https://pan.baidu.com/s/1DLQKdHk4BywWYf3jlR3PUw?pwd=tp3q

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
