# Peer review of "Linear Variation of M2 Tide in the East and South China Seas"

_EGUsphere, 2023_

## Author Comment (AC1)

Comments on 'Linear Variation of M2 Tide in the East and South China Seas' by Hao Ke et al. (Ocean Science)

This paper investigates long-term variations in the M2 component of the ocean tide in the waters near China using both coastal tide gauge and satellite altimeter data. The aim (although it is not very well expressed) is to see the extent to which the large changes in M2 found at the coast in tide gauge data are also found in the neighbouring waters in altimeter data. The authors claim to have devised a new method of tidal analysis in which a linear trend is combined with the usual harmonic terms.

Although the study is potentially of interest, I found the paper rather unsatisfactory in that the text is incomplete and confusing at times, and the authors do not demonstrate satisfactorily that the new tidal analysis method actually works. In particular, the discussion of the results towards the end of the paper is rather messy. To be fair, the latter is inevitable when the record lengths of the individual tide gauge and altimeter data sets are different. The assumption that any changes in M2 are linear in time is probably incorrect (or at least not demonstrated), given the variability in coastal and ocean processes, and so different record lengths will inevitably lead to different rates. (Showing linear rates of unequal length records is ok for general information purposes, of course.) Therefore, I am not sure what value the results would have, even if the data themselves had been perfect. I give some examples of these problems below.

A first comment is that the references in the introduction to the paper are inadequate. M2 is indeed known to be changing in parts of the world (the authors refer to Woodworth and Muller) but there is also a major recent review paper on that topic by Haigh et al. in 2019 (Prog. In Ocean.) which is not mentioned. And there is quite a quite a large literature on the subject with some papers in the Pacific (see the references in Haigh et al.). Also, they do not explain properly in the introduction how difficult it is to measure M2 accurately near to the coast using altimetry, see for example the review paper by Ray et al. in 2010 (Tide predictions in shelf and coastal waters, chapter 8 in S. Vignudelli et al., Coastal Altimetry, DOI: 10.1007/978-3-642-12796-0_8). So, they should explain that measuring changes in M2 near to the coast will be even more difficult than measuring M2 itself. Now, while a number of authors have tried to measure variations in M2 in the global ocean (Cherniawsky et al, CSR 2010 might be mentioned), I don't know of anyone who has looked seriously at changes near to the coast. Maybe I am wrong about this, but anyway the present authors don't properly explain what their aims are and what the status of previous work on this subject is.

First of all, thank you very much for your careful review and a lot of useful suggestions for helping us to improve the article. We have carefully read your comments one by one. When we read our article according to your suggestion, it is true that, as you said, the aim of the article is not clear, the references in the introduction are inadequate, the discussion of the results is confusion. Although time is limited, we try our best to revise the article. First, we want to elaborate what is the aim of this article. We hope to use the sea surface height observation from satellite to obtain the linear variation of main tidal constituent $M_2$

in China Seas. Because the time sampling rate of satellite altimetry is sparse, the method of harmonic analyzing year by year and then linear regression to obtain the linear change rate is obviously not applicable. So we tried the tide analysis method introduced in this article, and then constructed the $M_2$ linear variation model in China Seas by interpolation fitting method. In order to test the accuracy of the proposed tide analysis method, we have carried out statistics (mean difference and root mean square) at cross-over points of the ground tracks. As the same time we also want to compare with the results of the tide gauges, although the time ranges are not the same, mainly to provide a more variation information and reference. We revised the article and marked them in red. Please see the article for detail.

The revised introduction is as follows:

**1 Introduction**

Tidal movement is one of the main forms of dynamic processes in the ocean (Fang et al., 2004), and its motion is mainly represented by the harmonic constants of the tide. Tidal levels, tidal range and tidal currents vary on regular daily, fortnightly, monthly, annual, interannual, and longer-term time scales, driven by astronomical variations in the orbits and relative positions of the Sun, Moon, and Earth (Pugh and Woodworth, 2014). Perhaps because of stable and predictable of planetary orbital motions, tide predictions are reliability and tide constituents are consider to be stationary over time. Nonetheless, scientists and engineers have observed for some time that tidal amplitudes at many locations are shifting considerably due to nonastronomical factors, many of which remain poorly understood (Haigh et al., 2019).

For example, the $M_2$ tidal component is found to have significant seasonal periodic variation in the Arctic, the East China Sea and Yellow Sea, the Bay of Bengal, the northeast shelf of the Bering Sea, the Sea of Okhotsk and northern Australia (Mofjeld, 1986; Müller et al., 2014; Tazkia et al., 2016). Besides $M_2$, the amplitude and phase lag of other main tides also have the similar variations (Gräwe et al., 2014). Several major semidiurnal tides in the Bohai Sea of China have obvious seasonal variations (Fang, 1994). Based on two anchored water level observation data in Bohai Bay and using enhanced harmonic analysis (EHA) algorithm, it is found that the amplitudes and phase lag of the four main four components $M_2$, $S_2$, $K_1$ and $O_1$ show annual or semi-annual periodic changes, in which the relative variation of $M_2$ amplitude to the annual mean is nearly 10%. It is considered that these apparent tidal changes are caused by the seasonal variation of vertical eddy viscosity coefficient (Wang, et al., 2020). The method of EHA which can capture the harmonic parameters varying with time, is also applied to the northeast of the South China Sea, and found that $M_2$, $S_2$, $K_1$ and $O_1$ have irregular variations (Jin, et al., 2018). An analysis of water level time series from 20 tide gauges in Southeast Asia finds that diurnal and semidiurnal astronomical tides exhibit strong seasonal variability of both amplitude and phase, which is not caused by known modulations of the astronomical tide-generating forces (Devlin, et al., 2018).

In addition to the mentioned seasonal variation, there may be linear trends in tidal components. By analyzing the spatial trend distribution of the major tidal components, linear variations were found to be common along the North American coast (Woodworth, 2010; Müller, 2011). In addition, linear trends in the amplitude and phase of tidal constituents have been found around the British Isles, the southern North Sea, the Gulf of Maine, Northwest America, the North Atlantic Ocean, and the Caribbean Sea (Amin, 1983; Ray, 2006; Jay, 2009; Torres & Tsimplis, 2011; Feng & Tsimplis, 2014). Based on the tidal analysis of the water level data observed from 1954 to 2012 at 17 tidal gauges in the Bohai Sea, Yellow Sea, North Sea, and Taiwan Strait of China, the tidal constituents exhibit trends of long-term change. The largest increase is found for $M_2$ for which the amplitude increases by 4–7 mm/yr in the Yellow Sea, and the most significant rate is 7 mm/yr for the $M_2$ amplitude occurring at Lianyungang (Feng et al., 2015).

The Chinese coasts have been dramatically altered by sediment deposition in estuaries (Chu et al., 2006; Pelling et al., 2013), coastal erosion under the influence of frequent tropical cyclones (Qi et al., 2010; Feng et al., 2012), and extensive reclamation projects constructed over the past few decades, all of which may potentially alter the offshore tidal characteristics (Li et al., 2004; Li et al., 2013; Wang et al., 2014; Wang et al., 2011).

It appears that the linear variation in tidal amplitude and phase lag is objective and significant. However, most of the studies on the changes in tidal parameters have focused on the data obtained from tidal gauges along the continental coast, and the research on larger areas of offshore waters is rare. To address this, we used satellite altimetry data and tide gauges observation data to capture the changes in tidal parameters in the East and South China Seas. Moreover, considering the sparse time sampling rate of satellite altimetry data, the analysis method of tidal variation at tide gauges may not be applicable. Therefore, we proposed an additional time-varying model for the harmonic analysis of tide changes.

The remainder of this article is organized as follows. Section 2 provides a detailed description of the proposed method. The detailed introduction of the data used in this study are in Section 3. The variation of M2 derived from satellite altimetry and tide gauges, as well as the comparison and discussion of the results are arranged in Section 4. Finally, the conclusion and future research directions are presented in Section 5.

In lines 27-31, some numbers are quoted from Peng and Tsimplis and Peng et al. but where a trend value is given it should be clearer which one is referred to. One Peng paper used 17 stations and the other 20 for example.

Thanks. We have recalibrated the reference as the requested.

Line 32 might read better '… China, tidal constituents were found to exhibit ….'.

Thanks. We have modified according to the suggestion.

A second general comment is to do with the details of eq. (1) and following:

- I would denote mean sea level by Z instead of S, which can be confused with the sine terms following in eq. (3)

  Thanks. That's exactly correct what you said and we have changed $S(t)$ to $Z(t)$.

- It is usual to use V to represent the astronomical argument of a harmonic term, to which omega*time is added (omega being the speed). But the authors bundle the two together in their V which is unusual.

  I agree with your opinion. The change in phase of each tide constituent is the angular rate multiplied by time and plus the initial phase, which is the following expression, $V_0+\sigma t$. In fact, the second part $\sigma t$ can actually be written in terms of astronomical variables.

$$V = \mu_1\tau + \mu_2 s + \mu_3 h + \mu_4 p + \mu_5 N' + \mu_6 p' + \mu_0 \frac{\pi}{2}$$

  Where, $\mu_i$ is the Doodson number, and $\tau, s, h, P, N', p'$ are the 6 astronomical variables that varies with time. The angular rate of tide constituent is the derivative of V with respect to time $t$.

$$\sigma = \mu_1 \dot{\tau} + \mu_2 \dot{s} + \mu_3 \dot{h} + \mu_4 \dot{p} + \mu_5 \dot{N'} + \mu_6 \dot{p'}$$

  Therefore, $\sigma t$ can also be expressed by a linear combination of 6 astronomical variables. That is, the $\sigma t$ and initial phase $V_0$ can be uniformly represented by the phase V. We also made a supplementary explanation in the article.

- The various parameters are all said to be 'time-dependent', but that is misleading as it is surely only the amplitude (H) and phase lag (g) which are being considered as time dependent in the sense of the fitting. Of course, V, f and u are also time dependent but they are not actually time dependent free parameters. The f and u must have been assumed to have their equilibrium time dependence but the authors don't say. All this could be much clearer.

  Thanks. We agree with your opinion and we revised as you suggested. Please see article for details.

- I guess using just 13 harmonics is ok for present purposes but I have no idea what GB 12327-2022 is. It is not in the reference list.

  Sorry about that we didn't explain GB12327-2022. Actually, it is specifications for hydrographic survey issued by China in 2022. We also add this citations in the reference section.

- Line 48-49, this is not strictly true, in many parts of the world M2 exhibits a seasonal dependence of about 1% in amplitude – see for example the Pugh and Woodworth (2014) book (Cambridge Univ Press).

  Thanks. Actually, we quite agree with you. The tide constituent includes not only linear changes, but also seasonal or interannual periodic changes. As you said, our goal is to get a linear variation of the main tide M2 off the coast of China, and we pay more attention to that, so in the process of calculation, we don't take into account the periodic variation component, which may introduce error.

- One complication mentioned in Feng et al. concerns the fact that N2 will not be determined well with eq.1 because of its degree-3 component. That will have little effect on M2 but it could be mentioned for completeness.

  Thanks for your reminder. We also noticed that this sentence was mentioned in Feng's article:Note that where the $N_2$ constants linearly change the $M_2$ parameters also change. In fact, when we use the Rayleigh criterion to calculate the confluence period of $M_2$ and $N_2$, we find that the $M_2$ and $N_2$ can be completely separated. So we don't think $N_2$ will affect $M_2$, of course, this is just our view may not be correct.

- Line 56 – where

  Thanks. We have modified as your suggestion.

- Line 60 - the use of a notation of a dot over HC is confusing, I thought at first it referred just to H. I would put HC in brackets and make it clear that the dot refers to the whole bracket.

  Thanks. We have modified as your suggestion.

- In the matrix of eq.4 I would put the superscripts as subscripts as they could be confused with squared etc. Similarly in eq. 7, this reads to me as HC multiplied by delta-t, rather than being a function of delta-t, with the delta-t then squared. I know what the authors are doing here but it is not the best way of writing the algebra.

  Thanks. We have modified as your suggestion. Please see the article for the detail.

- m is the number of hourly values? (not mentioned). Also what is P in eq. 5?

  Yes, m is the number of observations of sea surface height. *P* is the weight matrix of the observations. In general, we approximate that satellite altimetry is equal weight observation, so *P* can be assumed to be a unit weight matrix. We also made a supplementary explanation in the article.

 Section 3 – why don't the authors use T/P data from 1992 which would provide a much longer record?

Because the time sampling rate of Jason satellite is very high, the main tidal constituents can be separated completely by tidal harmonic analysis method as long as the data length is more than 10 years. Jason series have accumulated more than 20 years of data from 1~3, which can fully meet the requirement. Another reason is that Jason and TP satellites have different missions and payloads, so their orbits are not exactly the same, and the accuracy of Jason satellite is better than that of TP. To sum up, we just only use the data of Jason 1~3.

Line 87 – exactly 35 days – the 'exactly' is important as the Envisat orbit is sun-synchronous

Yes, we quite agree with that the Envisat repeated period is exactly 35days, and in sun-synchronous with sun. 35 days is also integral number of the angular rate of the tide constituent S2. Therefore, the S2 cann't be isolated from Envisat observation data. We also carry on the explanation in the paper. Please see the article for detail.

Table 1 – there is no point giving the start/stop in seconds for this general information purpose, could not the format used in Table 2 be used here?

That's correct. Giving the start/stop in seconds doesn't make sense. We have removed the time and kept the date.

Line 95 – give references or web sites for UHSLC and the Hydrology Bureau. Also mention these data sources in the Acknowledgements.

Ok. We have added as requested.

Line 99 – 'can meet .. analysis'. This is obvious, you can do a tidal analysis on, say, a fortnight of data if necessary.

Thanks.

Figure 1 – please add Longitude (deg E) and Latitude (deg N) to the x-y annotation, add East and South China Seas to the map

Ok. We have added as requested.

Table 2 – deg E and deg N

Thanks. We have added according to your suggestion.

Section 3 has altimeter data mentioned before tide gauges, but the reverse in Section 4. Reverse the order in section 3.

Ok. We have modified as the requested.

Section 4 – the authors do not use the new fitting method here, but compute trends in M2 from those obtained from individual annual sets of data. Do the two methods give the same results? They don't say. That would then inspire confidence in its use with the altimeter data.

At the beginning, we considered that the tidal harmonic analysis were completed at the tide gauges and then linearly fitted the results of the tide constituents to get the change rate, which was simple to implement. We also hope that a similar process could be used to deal with satellite altimetry data, as you know, due to the low time sampling rate, it is impossible to make tidal harmonic analysis year by year like tide gauge. Therefore, we used the method mentioned in this paper. Actually, we have done experiments, and the results of the two methods are basically the same at the tide gauges.

Have subsection headings in section 4 as in section 3.

Yes. We have added the subsection headings in section 4.

Para at line 310 – more words of comparison to the results of Feng and Tsimplis and Feng et al. would be useful. If they differ, why so? Line 309 reword '… River, is the second longest, with an amplitude …'

Thanks. We have improved the comparison to the results of Feng and Tsimplis and Feng et al. Please see the article for detail.

Line 112 – basically ==> of the order of

Thanks. We have modified as your suggestion.

Line 113 – ditto

Thanks. We have modified as your suggestion.

Line 114 – this is a rather obvious statement. Say something like '… calculated at each point along-track' saying what the separation between points is.

Thanks. We have modified as your suggestion.

Line 115 – call it 'time varying term'. It doesn't seem to me there is a 'mode' here, just an extra linear term.

Yes, actually, it is just an extra linear term. Maybe there are some deviation in our understanding of mode. Actually, we prefer to use the word model instead, and we also changed all the mode to model.

How does the fitting method handle gaps in the records? Presumably for the annual tide gauge analyses, years of data were used only if they were, say, at least 80% complete.

In fact, the data we collected still meet 80% completion, whether they are satellite altimetry data or tide gauges observation data.

Figures 2-4 have a really poor colour scale with too much blue. Could not they be white at zero with pale to strong blue for negative values, and pale to strong red for positive ones (for example).

Thanks for your suggestion. We have modified the figures 2-4 as the request.

Fig 2 caption lines 1 and 2 – M2 amplitude and phase lag

Ok. We have added the caption as the request.

Fig 3 caption – ditto

Ok. We have added the caption as the request.

Line 124 – drop 'The following'.

Thanks for your suggestion. We have removed 'The following' in the article.

Line 131 – 'inconsistent values' is a strange expression. Say explicitly what is meant e.g. mean or absolute difference or root-mean-difference of M2 values at cross-over points of the ground tracks.

Sorry about confusion. The inconsistent values mean the differences at cross-over points of ground tracks. In fact, we just want to make a statistics on the differences at cross-over points, such as the mean and root-mean square for these differences. We revised these strange expression in the article. Please see the article for detail.

Table 4 – what is CR of M2? Does this mean rate of change of M2? Phase should be phase lag. 'Discrepancy Mean' should be 'Mean Difference'?

Yes. CR means the change rate of M2. The Phase is the phase lag. The Discrepancy Mean is the mean difference.

I think lines 135-170 need to be rewritten, perhaps with subheadings, to make it clearer what is being compared with what. There are 2 sorts of altimetry being compared with each other, and then altimetry with tide gauges, but the text is jumbled.

We have rewritten this part, please see the article for detail.

Line 139 – derived from ERS2/Envisat is much poorer.

Thanks for your reminder.

Drop 'The specific ..'

Thanks for your suggestion. We have removed 'The specific'

Line 145 – I can't see any dots in Fig 4 but I can in Figs 2 and 3. Those dots are not mentioned in the text.

It should be the color of the changing surface model that obscures these dots so that they are less obvious. These dots represents the locations of tide gauges along the coast, and their color represent the change rates. We have modified the colorbar to make the details clearer.

'Beyond the research scope'? What does that mean? Outside the gridded area?

Yes. It is outside the gridded area.

'Belongs to extrapolation' ==> 'has been obtained by extrapolation'

Thanks. We have modified according to your suggestion.

Line 152 – I think should read 'Comparing the rates of change of M2 obtained at the tide gauges and altimetry at tide gauge locations …'

Yes. We have modified.

What are the standard deviations'? You mean the root mean squares?

Yes. We use standard deviations for root mean squares.

Line 154 – 'crossing points in general'

Thanks. We have modified as your suggestion.

Is Table 5 using just Jason data or the combined altimetry mapping? Actually I thought these values were quite interesting – what is the correlation between the sets of trend-differences?

The data in the table 5 are the comparison between the combining altimetry mapping and the tide gauges. Actually, we didn't find any correlation between these differences. We believe that these differences can be related to the accuracy of satellite altimetry and the data quality of tide gauges. In principle, the better the accuracy of these observations or tide gauge's data, the smaller the difference. Of course, there is a premise that the time periods of the two data should be the same.

Line 159 – 'According to ..'. But MS4 is not actually listed in Table 2. I agree MS4 could be a complication in the case of Envisat.

Thanks for your reminder. What we mean is that all tide gauges in Table 2 are completed tidal harmonic analysis, and found that their shallow tide constituent MS4 had a small order of magnitude in amplitude, about a few centimeters.

Line 161 – 'to be poorer'

Thanks for your reminder.

Line 164 – by mainland do you mean the Chinese mainland and not Taiwan or do you mean the coast in general? Reword.

Sorry about confusion. We mainly express that the tide gauges are generally located in the coast where the accuracy of altimetry is poor.

As mentioned, I thought Table 5 interesting but does this use just Jason altimetry or combined? Also what are the modes? – you mean just the gridded altimetry data sets. Why is the mean of the absolute difference used here whereas mean difference is used in Table 4?

We combined the variation results of Jason series and Envisat series. Actually, the word mode just means the constructed change rate model, which is a function of the coordinates.
After revision, the Table 4 in the original article has become Table 3. In fact, the differences in Table 3 and 5 are both absolute values. We are sorry that it is our failure to state clearly at first that lead to confusion.

Lines 173-on and Figure 4 – I thought Figure 4 was potentially quite interesting, with large changes off-shore, although I suspect some of that is due to noise in the altimetry mapping.   It seems to me that, if the paper had been written first as a mapping exercise for M2 trend from altimetry, then it would have read much better. As it stands there is a mixture being discussed in the paper of that mapping using data from different missions, and also consideration of what is happening at the coast at the tide gauges compared to the altimetry. And, as I mentioned above, the different spans of data do not make for an easy discussion. The speculation that any tidal changes are due to water depths and coastline changes is probably correct, but speculation it remains, there is no research on that presented here. Maybe if the text of the paper was revised then some of these objections could be removed.

---

## Author Comment (AC2)

The paper examines tide gauge data and satellite altimeter data in the East and China Seas in an attempt to determine possible linear trends for the M2 ocean tide.   That topic is timely, but unfortunately I do not find the quality of this work high enough to meet the standards of Ocean Science. The analysis is pedestrian (at best), the English rather poor, and the discussion superficial.   Below I briefly mention major shortcomings; I do not bother tabulating minor problems. My overall recommendation is that OS rejects the paper.

1. I had hope that a discussion of tide-gauge trends around China might bring forth new data. Alas, the authors use the standard set of old data archived at the University of Hawaii which has not been updated since 1997. (The exceptions are two gauges from Taiwan, which are recent, and Hong Kong and one other site, mentioned below.) Evidently the government of China continues to withhold tide gauge data from scientific study.   For this reason, there is nothing new in the tide gauge analysis that was not already published by Feng, Tsimplis, and Woodworth (2015). Moreover, the analysis and results of Feng were superior and more complete than the work reported here.

On this same point of missing modern data: The authors emphasize the large amount of coastal changes now ongoing along the Chinese coast. And yet they use tide-gauge time series that stop 25 years ago. It seems more modern data, and not just from Taiwan, are required to examine properly this problem.

In fact, we mainly want to use satellite altimetry data (Jason and Envisat) to calculate the variation of tidal harmonic constants. The data of tide gauges along the coast are mainly used to verify the results of satellite altimetry solution, although the time ranges of both data collections are different. It has no choice but using these data derived from University of Hawaii, especially since we just collected only one tide gauge Lianxinggang. In addition, the harmonic analysis method combined with an additional time-varying model in this paper is not exactly the same as that used by Feng, Tsimplis and Woodworth. Moreover, what we focus on is China's open ocean, rather than the several places where the tide gauges are located.

We have modified the introduction and discussion of the article, and the language has been re-polished. Please review.

2. There is not a single error bar in the entire paper. There is no way to know if any of the computed trends are significant or not.   Error analysis (confidence intervals) are critical to this kind of work. Moreover, a proper error analysis is not trivial -- i.e., one cannot just use uncertainties given by some regression package that assumes white noise. Thus, this critical part of the work is missing.

Yes, we quite agree with you. We have added 95% confidence interval to the calculation results of the tide gauges in table 4. Please review.

3. There is no appreciation for the fact that 18.6-year nodal modulations of tides can potentially impact estimation of tidal trends, if the modulations are non-equilibrium. In fact, an important result from Feng et al. (2015) is that, indeed, the nodal modulations at many of these stations are non-equilibrium. The authors should have considered this point, since it was stressed by Feng et al.

Thanks a lot for your advice. Indeed, the 18.6-year nodal modulations is likely to affect the estimation of the linear variation of M2 tide. In the conclusion of Feng's article, the observed modulations of the M2 and N2 amplitudes are smaller than theoretically predicted at the northern stations and larger at the southern stations. Therefore, this critical part should be taken into account in our discussion. We have modified according to your suggestion and Feng's conclusion. Please review.

4. There is no reference to the work by Bij de Vaate et al. (doi: 10.1029/2022JC018845 ), who made a much more thorough study of possible tidal trends from satellite altimetry. They reported trends (with uncertainty analysis) in the East China Sea, which should have been compared here. They also found that trends in the South China Sea were not significant, at least at satellite cross-over locations.

Thanks a lot for your advice. It is our fault to neglect this excellent work by Bij de Vaate et al. Their work showed the change rate the main tide constituents in the global sea. Due to the resolution, we can roughly see that the change rate of M2 is quite small in the China Seas, especially in the South China Sea. We have introduced this work in the introduction of the article. Please review the article for detail.

5. There is no appreciation for the possible existence of systematic errors in satellite altimetry which could impact trend estimation. Nor is there discussion of the apparently large errors in M2 (not M2 trends) seen at Jason cross-overs -- 12.8 cm, according to Table 4.   It is difficult to see how mm/year trends in M2 could be determined in the presence of such large noise in the mean M2.

Yes, we agree with you that there may be possible existence of systematic errors in satellite altimetry which could impact the M2 trend estimation. In the South China Sea, the RMS of M2 amplitudes at Jason cross-overs is about 12.8 cm, which is of a large order of magnitude. Therefore, we also agree with what you mentioned in point 2 to add error bar. The parameters change rates of M2 is quite small, so the system error is likely to cover the change rate. However, we also hold another view that even if there is a systematic error in it, we calculate the rate of change, the systematic error may not have much effect on the rate of change, because in calculating process, the systematic error may be eliminated by the first difference. Therefore, when calculating the amplitude, phase lag and the corresponding change rate of M2, the influence of systematic error on the former may be more intense. We have added a discussion of the results in table 4. Please review the article for details.

6. Much of the mathematics laid out, especially the large matrix in Eq (4), is not needed. Everybody already knows how to set up a least-squares problem.

Actually, we used the tidal harmonic analysis model of time-varying analysis, which is different from the traditional tide harmonic analysis. We expanded the sine and cosine part of tide constituent into a linear combination, and incorporate the linear change rate and the initial term into the least-square calculation. So we hope to show the detailed calculation process.

7. In both the Abstract and the Conclusions, it is stated that the detected tidal trends in M2 are caused by changes in water depth and coastlines of estuaries. There is no evidence presented that backs up these statements. They are merely assertions.

Indeed, there is no direct evidence to prove the tidal parameters changes caused by the water depth and coastlines of estuaries. Therefore, we just stated that the rapid change in water depth may be one cause. Actually, there are also related literatures that through tide motion numerical experiments, and found that the coastline and water depth variation could significantly alter $M_2$ amplitude in the Bohai Sea and Yellow Sea (Pelling et al., 2013). In recent years, especially in the past two decades, great changes have taken place in many coastal areas in China, including the depth and coastline. It is on these grounds that we have made this speculation. Of course, we are also planning to carry on relevant experiments, hoping to prove it.

8. As far as I can tell, there is one tide gauge used here that is not from the UHSLC: the gauge at Lianxinggang. In the "Data Availability" statement, the authors give a web site for these data, but the link did not work for me.

Sorry about that, we upload the tide gauges data on the Baidu network disk, and the foreign IP address may not be able to access China's domestic links. We will re-upload the data later, putting them on a publicly downloadable network disk, or submit it directly to the editorial department.

9. I had difficulty with the color scale used in Figures 3 and 4. It is not easy to distinguish positive trends from negative trends, let alone decipher the magnitudes.

Yes, this point was also mentioned in the comments from the reviewer 1, and we have revised the color scale according to the request. Please review.